# Effect of *Hanseniaspora uvarum*–*Saccharomyces cerevisiae* Mixed Fermentation on Aroma Characteristics of *Rosa roxburghii* Tratt, Blueberry, and Plum Wines

**DOI:** 10.3390/molecules27228097

**Published:** 2022-11-21

**Authors:** Mingzheng Huang, Xiaozhu Liu, Xin Li, Xiaofang Sheng, Tingting Li, Weiyuan Tang, Zhihai Yu, Yuanmeng Wang

**Affiliations:** 1Guizhou Institute of Technology, College of Food & Pharmaceutical Engineering, Guiyang 550003, China; 2Key Laboratory of Microbial Resources Collection and Preservation and Rural Affairs, Beijing 100081, China; 3School of Public Health, Guizhou Medical University, Guiyang 550025, China; 4College of Liquor and Food Engineering, Guizhou University, Guiyang 550025, China

**Keywords:** *Hanseniaspora uvarum*, fruit wine, aroma, *Rosa roxburghii* Tratt, blueberry, plum

## Abstract

*Hanseniaspora uvarum*, a non-*Saccharomyces cerevisiae* species, has a crucial effect on the aroma characteristics of fruit wines, thus, attracting significant research interest in recent years. In this study, *H. uvarum*–*Saccharomyces cerevisiae* mixed fermentation was used to ferment *Rosa roxburghii* Tratt, blueberry fruit wine, and plum fruit wines using either a co-inoculated or a sequentially inoculated approach. The three fruit wines’ volatile aroma characteristics were analyzed by headspace–solid-phase microextraction–gas chromatography-mass spectrometry (HS–SPME–GC–MS). The results showed that the mixed inoculation of *H. uvarum* and *S. cerevisiae* reduced the alcoholic content of Kongxinli fruit wine. Moreover, *H. uvarum*–*S. cerevisiae* fermented *Rosa roxburghii* Tratt, blueberry, and plum fruit wines and further enriched their flavor compounds. The overall flavor characteristics of sequentially inoculated fruit wines differed significantly from those fermented with *S. cerevisiae* alone, although several similarities were also observed. Sequential inoculation of *H. uvarum* and *S. cerevisiae* positively affected the mellowness of the wine and achieved a better harmony of the overall wine flavors. Therefore, *H. uvarum*–*Saccharomyces cerevisiae* mixed fermentation can improve the complexity of the wines’ aromatic composition and empower them with a unique identity. In particular, *H. uvarum*–*Saccharomyces cerevisiae* blueberry wine produced by mixed fermentation had the widest variety and content of aroma compounds among the fermented wines. Therefore, *H. uvarum*–*Saccharomyces cerevisiae* mixed-fermentation inoculation in the three fermented fruit wines significantly increased the aroma compound variety and content, thus, enriching their aroma richness and complexity. This study is the first comparative evaluation of the aroma characteristics of different fruit wines fermented with a mixed inoculation of *H. uvarum* and *S. cerevisiae* and provides a preliminary guide for these fruit wines produced with non-*Saccharomyces* yeast.

## 1. Introduction

China is the world’s largest producer of fruits [1]. Thus, improving the deep processing of fruits with processes such as winemaking can increase their edible added value, better address fruit production–marketing imbalance, and promote the sustainable development of the fruit industry [2,3]. However, fruits are perishable due to their short shelf life. Fruits are the primary raw material for brewing fruit wine, and various types are used to produce popular beverage fruit wines due to their unique flavors [4,5].

*Hanseniaspora uvarum*, a subspecies of *Saccharomyces apiculatus*, is present in various natural substances, including different fruits and their fermented products [6,7]. *H. uvarum* is closely related to wines’ aroma (odor) characteristics [8,9,10]. *H. uvarum*–*S. cerevisiae* mixed fermentation improved the content of typical characteristic aroma components—for example, the floral and rosé characteristics of rosé wines—and accentuated the varietal characteristics of rosé wines [11]. In addition, *H. uvarum*–*S. cerevisiae*, as a mixed fermenting agent, improves the sensory wine quality and reduces volatile acids’ content [12]. However, research on the fermentation characteristics of *H. uvarum* has primarily focused on the grapevine winemaking sector [13,14], and its effects on the aroma characteristics of wines from other fruit lack evidence.

The improvement of fruit wine aroma compounds by mixed multi-strain fermentation is increasingly used for the fermentation production of a wide range of fruit wines [15,16]. Indeed, all biochemical reactions are carried out during co-fermentation by multiple microorganisms. *S. cerevisiae* has long been used for wine production due to its high fermentative power and alcohol productivity [17]. In contrast, non-*Saccharomyces cerevisiae* species have a lower fermentation rate. Still, they release various enzymes during the wine fermentation process that metabolically break down the aroma precursors in the raw materials and promote the release of aromas, resulting in a richer wine flavor [18].

In our previous study, we evaluated the oenological properties of three strains of *H. uvarum* and found that the selected *H. uvarum* strains increased the volatile aroma richness and complexity of *Rosa roxburghii* Tratt (*R. roxburghii*) wine by fermentation with *S. cerevisiae* [19]. In this study, fresh *R. roxburghii*, blueberry, and plum were used as the main brewing ingredients, while *H. uvarum* mixed with *S. cerevisiae* was used as a fermentation agent to further compare the effects of *H. uvarum* on the aroma characteristics of different fruit wines. Subsequently, headspace solid-phase microextraction (HS–SPME) combined with gas chromatography–mass spectrometry (GC–MS) was used to analyze the different fermentation methods’ effects on the volatile aroma components in fruit wines. Finally, the volatile components’ contribution to the different fruit wine aromas was estimated using odor active value (OAV). Overall, our findings provide a scientific basis for further improving the fruit wine brewing processes.

## 2. Results and Discussion

### 2.1. Effect of H. uvarum–Saccharomyces cerevisiae Mixed Fermentation on the Alcoholic Content of Different Fruit Wines

The alcoholic content of the three fruit wines produced by *Hanseniaspora uvarum*–*Saccharomyces cerevisiae* mixed fermentation is presented in Table 1. The five fermentation groups of *R. roxburghii* fruit wines had relatively similar alcoholic content. However, in both blueberry and plum fruit wines, significant differences in the alcoholic content were observed between the different mixed-fermentation methods. In the blueberry wine, the highest alcohol content was recorded in the commercial *S. cerevisiae* fermentation treatment, followed by the co-inoculated *H. uvarum* F119, *H. uvarum* 32349 treatment, and the sequentially-inoculated *H. uvarum* treatment. In fermented plum wine, the sequentially inoculated *H. uvarum* F119 treatment had the same alcoholic content as the commercial *S. cerevisiae* fermentation. The co-inoculated *H. uvarum* 32349 treatment had the lowest alcohol content. Therefore, *S. cerevisiae* inoculation did not affect the alcoholic content of *R. roxburghii* wine but reduced the blueberry wine alcoholic content. In addition, the sequential inoculation with *H. uvarum* and *S. cerevisiae* did not affect the plum wine’s alcoholic content. However, co-inoculation with *H. uvarum* and *S. cerevisiae* reduced the alcoholic content of plum wine.

### 2.2. Effect of H. uvarum–Saccharomyces cerevisiae Mixed Fermentation on the Fruit Wines’ Volatile Aroma Components

#### 2.2.1. Effect of *H. uvarum*–*Saccharomyces* cerevisiae Mixed Fermentation on the Types of Volatile Aroma Compounds in Different Fruit Wines

The aroma compounds in fruit wines, mainly produced by alcoholic fermentation through yeast, have a prominent contribution to the quality characteristics of fruit wines [20]. Thus, the aroma compounds content is of great significance for the quality of fruit wines and is a key indicator of their quality. The aroma compounds in the three fruit wines were detected and analyzed using HS–SPME–GC–MS. Six major aroma compound classes were detected, including esters, alcohols, acids, aromatics, aldoketones, and others. In total, 48 aroma compounds were detected in the *R. roxburghii* juice before fermentation. In addition, 62 (*S. cerevisiae* X16), 66 (*H. uvarum* F119-G), 64 (*H. uvarum* F119-S), 62 (*H. uvarum* 32349-G), and 57 (*H. uvarum* 32349-S) volatile aroma chemicals were detected in *R. roxburghii* fruit wine under the respective fermentations (Figure 1A). Moreover, 65 volatile aroma compounds were detected in blueberry juice. Fifty-five (*S. cerevisiae* X16), 58 (*H. uvarum* F119-G), 62 (*H. uvarum* F119-S), 57 (*H. uvarum* 32349-G), and 56 (*H. uvarum* 32349-S) volatile aroma compounds were detected (Figure 1B) in blueberry wine. Finally, 34 volatile aroma compounds were detected in plum juice. An amount of 37, 51, 50, 52, and 51 volatile aroma compounds were detected in the *S. cerevisiae* X16, *H. uvarum* F119-G, *H. uvarum* F119-S, *H. uvarum* 32349-G, and *H. uvarum* 32349-S fermentation treatments, respectively (Figure 1C). In addition, the yeast-fermentation capacity increased the number of aroma compounds, predominantly esters, in the fruit wines. As a result, *H. uvarum*–*Saccharomyces cerevisiae* mixed fermentation increased the variety of aroma compounds and the richness of *R. roxburghii* and plum fruit wines.

#### 2.2.2. Effect of *H. uvarum–Saccharomyces Cerevisiae* Mixed Fermentation on The Content of Volatile Aroma Compounds in Different Fruit Wines

The aroma compound content in the three fruit juices and their respective fruit wines was further analyzed (Table 2). In total, 30.88 mg/L of aroma compounds was detected in the *R. roxburghii* juice. The aroma compound content in the five fermentation treatments of *R. roxburghii* fruit wines increased significantly, with the lowest content measured in the *S. cerevisiae* X16 group and the highest content in the *H. uvarum* 32349-S group (1948.69 mg/L for *S. cerevisiae* X16, 2086.45 mg/L for *H. uvarum* F119-G, 2130.04 mg/L for *H. uvarum* F119-S, 2011.92 mg/L for *H. uvarum* 32349-G, and 2403.16 mg/L for *H. uvarum* 32349-S). There were no statistically significant differences in the total content of aroma compounds among the five *R. roxburghii* fruit wines. However, there were certain differences in specific aroma compound content among different yeast-fermented *R. roxburghii* fruit wines. The highest ester content was found in the *H. uvarum* 32349-S-fermented *R. roxburghii* fruit wine group, whereas the lowest was found in the group with *S. cerevisiae* X16 alone. In addition, the ester content of the remaining three groups of *R. roxburghii* fruit wine was lower than that of the *H. uvarum* 32349-S fermentation and higher than *S. cerevisiae* X16 fermentation. On the contrary, the highest alcohol content was measured in the *S. cerevisiae* X16 single fermented *R. roxburghii* fruit wine, followed by the *H. uvarum* 32349-S-fermented group, which were both significantly higher than the remaining three *R. roxburghii* wine fermentations.

*S. cerevisiae* fermentation in the blueberry fruit wine significantly increased the content of esters, alcohols, acids, aromatics, and other compound classes, while it decreased the aldoketone content compared to its unfermented juice (Table 3). The ester content in the blueberry fruit wines was relatively similar and did not significantly differ among the five fermentation groups. Alcohol content ranged from 212.03 to 333.37 mg/L. The ethanol content was relatively similar among the *S. cerevisiae* X16, *H. uvarum* F119-G, and *H. uvarum* 32349-G fermentations. The highest content of acids was measured in the *H. uvarum* F119-S and the lowest in the *H. uvarum* 32349-G fermentation, respectively. Overall, yeast fermentation significantly reduced the aldoketone content, while it increased the content of aromatics and other compounds in blueberry fruit wine.

The total aroma compounds measured in the plum juice were only 11.52 mg/L. In the plum fruit wine, however, they increased significantly to 2884.44 mg/L (*S. cerevisiae* X16), 3425.86 mg/L (*H. uvarum* F119-G), 3980.39 mg/L (*H. uvarum* F119-S), 3281.48 mg/L (*H. uvarum* 32349-G), and 4143.48 mg/L (*H. uvarum* 32349-S), respectively. Yeast fermentation significantly increased the content of esters, alcohols, acids, aromatics, and other compound classes, while it significantly decreased aldoketone content. Notably, no aldoketones were detected in all five fermentation treatments of plum fruit wines (Table 4).

#### 2.2.3. *H. uvarum–Saccharomyces cerevisiae* Mixed-Fermentation Effect on Volatile Esters and Alcohols in Different Fruit Wines

Esters in fruit wines, produced primarily by fruit and alcohol fermentation, have a fruity aroma. Thus, they greatly influence the wine aroma profile [21]. A total of 41 esters, mostly caproates and caproate compounds, were detected in the *R. roxburghii* fruit wines produced by the different fermentation methods (Table 2). The ester compounds content in each fermentation treatment reached more than 80%, except for the *S. cerevisiae* X16 fermentation. Blueberry fruit wine had 26 ester compounds, which increased the relative content by 11.45% compared to blueberry juice. In addition, the ester content in the *H. uvarum*–*Saccharomyces cerevisiae* fruit wine produced by mixed fermentation exceeded 64.5%, indicating that *H. uvarum*–*Saccharomyces cerevisiae* mixed fermentation positively affected the production of esters. Esters were composed of higher levels of ethyl esters, up to 3,496,094.31 μg/L in fruit wines. Furthermore, the ethyl esters content in sequentially fermented fruit wines was higher than that of the co-fermented wines. The higher levels of ethyl esters gave the fruit wines a pleasant cheesy, fruity flavor, making them more fragrant.

Alcohols are metabolites with mellow and fruity aromas formed by yeast, either through sugar catabolism or decarboxylation reactions and amino acid deamidation, which can impart a richer and more intense aroma to fermented fruit wines [22]. An amount of 13, 20, and 13 alcohols were detected in *R. roxburghii*, blueberry, and plum fruit wines across all the fermentation methods. In the *R. roxburghii* and blueberry fruit wines, the highest concentrations of alcohols were measured in the single *S. cerevisiae* X16 fermentation treatment. In contrast, their concentrations in the wines produced by the other fermentation methods were lower (Table 2 and Table 3). In the plum fruit wine, the highest alcohol content was recorded in the *H. uvarum* 32349–*Saccharomyces cerevisiae* mixed-fermentation groups (Table 4). Isobutanol, isoamyl alcohol, hexanol, and 2,3-butanediol were the major alcohols detected across all three fruit wines and all fermentation methods. Among them, isobutanol and 2,3-butanediol were the alcohols produced after fermentation, providing the wines with fruity and sweet aromas. Isoamyl alcohol was present at high levels in all three fermented fruit wines, with content greater than 25,000 μg/L in the blueberry and plum fruit wines. Isoamyl alcohol, the major component of fusel oil or higher alcohols, can be synthesized by *Saccharomycetes* during fermentation through the amino acid anabolic pathway [23]. In the present study, the higher alcohols content in the three *H. uvarum*–*Saccharomyces cerevisiae* fruit wines produced by mixed fermentation may be attributed to the gradual decay of *H. uvarum*, as ethanol concentration increases in the late fermentation stage. As a result, *H. uvarum* is absorbed and utilized by *S. cerevisiae* as a nutrient source after *H. uvarum* decomposition [24]. Higher levels of alcohols were detected in all three fruit wines. The higher alcohol content in the volatile compounds of *H. uvarum*–*Saccharomyces cerevisiae* fruit wines produced by mixed fermentation ranged from 150 μg/L to 55,000 μg/L. Higher alcohol concentrations below 300,000 μg/L impart a pleasant style to the wine [25]. Therefore, *H. uvarum* 32349–*Saccharomyces cerevisiae* mixed fermentation can be beneficial in fine-tuning the aroma characteristics of fruit wines.

#### 2.2.4. *H. uvarum–Saccharomyces cerevisiae* Mixed-Fermentation Effect on Volatile Acids, Aldoketones, and Other Compounds in Different Fruit Wines

Fermented fruit wines contain acids, aldoketones, aromatics, and other compounds, in addition to esters and alcohols. Acids provide milk and cheese flavors at low concentrations, while when present at very high concentrations they produce putrid and sour flavors [26]. *H. uvarum*–*Saccharomyces cerevisiae* mixed fermentation increased the acid content in fruit wines. However, the resulting differences between *H. uvarum*–*Saccharomyces cerevisiae* mixed fermentation and single *S. cerevisiae* fermentation were not significant (Table 2, Table 3 and Table 4). Several fatty acids, including capric acid, hexanoic acid, and octanoic acid, had a higher concentration in the mixedfermented fruit wines than in single-fermented wines, indicating that *H. uvarum*–*Saccharomyces cerevisiae* mixed fermentation contributed to the production of fatty acids and other acids in the wine. Aromatic compounds were also present in most fruit wine fermentations. They were slightly higher in content in sequentially fermented fruit wines than in co-fermented fruit wines, especially in *H. uvarum* 32349-S-fermented blueberry wines (reaching 15.40%). Most aromatic compounds provide floral and fruity aromas, improving wine quality. 2,6-butylated hydroxytoluene was significantly higher (395,891.35 μg/L) in the sequentially fermented blueberry wines, providing a fresh and cool aroma. Compared with the SC-fermented fruit wines, HSMF-produced blueberry fruit wines had a higher capacity to produce aromatic compounds. Therefore, HSMF promoted the aroma composition of blueberry fruit wines. Aldoketones, which are produced by alcohol oxidation, can provide fruit aromas to fruit wines. However, these compounds are unstable and can be further oxidized to carboxylic acids, which reduces their content. Several compounds, such as furfural and 5-hydroxymethylfurfural—which were only detected in the *H. uvarum* 32349–*S. cerevisiae* X16 mixed-fermented *R. roxburghii* wine—impart sweet, roasted, and caramel flavors to fruit wines. The total aldoketone content was higher in the HU 32349–*S. cerevisiae* X16 mixed-fermented wines than in other fermentation treatments. However, the difference in the total aldoketone content between the *H. uvarum* 32349*–S. cerevisiae* X16 mixed fermentation and the single fermentations was not significant. The three fruit wines also contained other volatile substances besides the above compounds. 2,2-dimethyl-5-methylenebicyclo [2.2.1] heptane was only detected in the mixed-fermented *R. roxburghii* fruit wine. Linalool oxide was detected at certain levels in blueberry fruit wine. Furthermore, 2-methyl-1,5-dioxaspiro[5.5]undecane gave the plum wine a herbal flavor. Overall, the above compounds provided the fruit wines with a distinctive aroma profile.

### 2.3. Heat Map Analysis of Volatile Components in H. Uvarum–Saccharomyces Cerevisiae Mixed-Fermentation Produced Fruit Wines

A heat map was constructed to analyze the differences in volatile composition among *R. roxburghii*, blueberry, and plum fruit wines across the different fermentation methods used. The heat map depicts the general volatile profile through a color gradient in the raw *R. roxburghii* juice and the corresponding *R. roxburghii* wine produced by different fermentation methods. The normalized color scale is from 0 (blue) to 1 (red corresponds to the abundance of volatiles from low to high) (Figure 2). The aroma composition of the sequentially inoculated fermented *R. roxburghii* fruit wine was similar to the *S. cerevisiae* fermented wine, indicating that different mixed fermentation strategies resulted in different aroma profiles of the fruit wines. The aroma profile differences between the *H. uvarum* 32349 mixed fermentation and the *H. uvarum* F119 mixed fermentation were not significant. In addition, the co-fermentation treatments of both non-*Saccharomyces cerevisiae* yeasts were clustered together. Similarly, the sequentially fermented treatments formed a distinct cluster

A significant difference in volatile compound content was observed between raw *R. roxburghii* juice and its corresponding wine. In addition, there were substantial differences in the volatile compounds of *R. roxburghii* wines produced by various fermentation methods, both in volatile type and composition. Various esters were produced after fermentation, while several esters detected prior were undetectable after fermentation. For example, 1-ethylpropyl acetate, 2-pentyl acetate, (Z)-2-pentenyl acetate, 2-methylbutyl acetate, (Z)-2-hexenyl acetate, 3-cyclohexenyl acetate, and 2,3-butanediol diacetate were detected in high concentrations in the raw *R. roxburghii* juice, however, they were not present in the five fermented *R. roxburghii* wines after fermentation. The HSMF groups (32349G, 32349S, F119G, F119S) exhibited a higher ester content than the single SC fermentation group (X16). Octyl acetate was only detected in the *H. uvarum* F119 co-fermented *R. roxburghii* wine at a relatively high concentration. Alcohols increased significantly after fermentation, except for 1-penten-3-ol, (E)-2-hexenol, and octanol. In addition, 2-hydroxy-γ-butyrolactone, 2-hydroxy-γ-butyrolactone, furfural, and 5-methylfurfural were detected and showed an increasing trend only in the HU 32349 mixed-fermented *R. roxburghii* wine compared to the unfermented raw *R. roxburghii* juice. Geranyl acetate and 3-furanmethanol were only detected in the *H. uvarum* 32349*–S. cerevisiae* co-fermented *R. roxburghii se* wine. Eight volatiles were only detected in the *H. uvarum* F119-fermented *R. roxburghii* wines. Octyl acetate, nerol acetate, and 1-methyl naphthalene were only detected in the *H. uvarum* F119-SC F119G co-fermented *R. roxburghii* wines. Furthermore, isothiocyanate cyclohexane and 3-(methylthio)-1 propanol/methionol were only observed in the *H. uvarum* F119-fermented and the *S. cerevisiae* X16 sequentially fermented *R. roxburghii* wines. Finally, ethyl benzoate and butyl benzoate were detected in the *H. uvarum* F119 co-fermented and sequentially fermented *R. roxburghii* wines.

Similarly, the volatile compound content was higher in mixed-fermented blueberry fruit wines than in the single-fermented ones. The concentration of alcohols was low, while esters and aromatics were significantly higher in concentration in the mixed-fermented wines than in single-fermented wines (Figure 2B). The mixed-fermented plum wine volatile composition was considerably richer than that of the corresponding single-fermented wine (Figure 2C). In the mixed-fermented plum wine, more volatiles were observed in the *H. uvarum* F119-fermented wine than in *H. uvarum* 32349. Isopentyl dodecanoate, (E)-3-hexenyl acetate, isobutyl hexanoate, and isopentyl dodecanoate were only detected in the mixed-fermented plum wines. Among them, (E)-3-hexenyl acetate was only detected in the *H. uvarum* 32349-fermented wine, isopentyl dodecanoate in the *H. uvarum* F119-fermented wine, isopentyl butanoate and isobutyl hexanoate in both *H. uvarum* 32349- and *H. uvarum* F119 co-fermented wines, and nonyl acetate and 2,3-butanediol in the *H. uvarum* 32349 sequentially-fermented plum wine, respectively.

### 2.4. PCA of GSNF-Produced Fruit Wines

A PCA model was constructed from the volatile aroma compound compositions of three fruit wines produced by different fermentation methods to investigate the similarities and differences in volatile profiles among these fruit wines. The hexagons of each color in the score plot represent the *R. roxburghii* and blueberry wines produced by different fermentation methods. The green circles in the loadings diagram represent the individual volatiles. The volatile aroma profiles differed significantly between the different fruit juices and the corresponding fruit wines produced by different fermentation methods (Figure 3). In the score plot, a closer distance between the samples indicates a more similar aroma profile, while samples distanced apart from each other have a more distinct aroma profile. In addition, a separation was observed between the fruit juices and the fruit wines produced by different fermentation methods. Most of the fruit wines were distinctly dispersed in the PCA. Among them, *H. uvarum* 32349-G, F119-G, and *S. cerevisiae* X16 were clustered together, whereas 32349-S and F119-S formed a distinct cluster. This indicated that 32349-G, F119G, and X16-fermented *R. roxburghii* wines had high similarity in volatile compound composition, while 32349-S was similar to F119-S. In addition, more volatile aroma compounds were distributed around the fermented fruit wines in the loadings diagram. Therefore, the variety of volatile aroma components in *R. roxburghii*, blueberry and plum wines produced significantly increased after fermentation. Although the fruit wine aroma compound content decreased in both the co-fermented and sequentially fermented treatments, a difference was observed between them.

The PCA plot of fruit wines produced by different fermentation methods suggested that 2-hydroxy-γ-butyrolactone (38), furfural (75), 5-methylfurfural (76), geranyl acetate (34), and 3-furanmethanol (57) were associated with the *H. uvarum* 32349 mixed-fermented *R. roxburghii* wine (Figure 3A). Geranyl acetate (34) and 3-furanmethanol (57) were only detected in *H. uvarum* 32349–*S. cerevisiae* mixed-fermented *R. roxburghii* wine. Octyl acetate (16), nerol acetate (33), and 1-methyl naphthalene (69) were more abundant in the *H. uvarum* F119-SC F119G mixed-fermented *R. roxburghii* wine. Isothiocyanato cyclohexane (39) and 3-(methylthio)-1-propanol/methionol (55) were abundant in the *H. uvarum* F119S mixed-fermented *R. roxburghii* wine. Interestingly, the volatile compounds of blueberry wine and *R. roxburghii* wine exhibited similar trends. The volatile compound content was higher in mixed-fermented blueberry wines than in single-fermented blueberry wines. However, the content of alcohols was low, and esters and aromatics were significantly higher in the mixed-fermented wines compared to the single-fermented wines (Figure 3B). The volatile compounds distributed near the mixed-fermented plum wine were higher in content than those near the single-fermented wine. In the mixed-fermented plum wine, more volatiles were detected in the *H. uvarum* F119 fermentation than in the *H. uvarum* 32349 (Figure 3C). Isopentyl butanoate (11), (E)-3-hexenyl acetate (9), isobutyl hexanoate (15), and isopentyl dodecanoate (34) were co-localized in the PCA with the mixed-fermented plum wine. Among them, (E)-3-hexenyl acetate (9) was highly correlated with *H. uvarum* 32349 mixed-fermented *R. roxburghii* wine. Isopentyl dodecanoate was only detected in the *H. uvarum* F119 mixed-fermented plum wine. In addition, nonyl acetate (23) and 2,3-butanediol (45) were associated, according to the loadings plot, with the *H. uvarum* 32349 sequentially fermented (32349S) plum wine. The volatile components of the three fruit wines produced by different fermentation methods exhibited similar trends. Notably, the mixed-fermented wines were richer in volatiles than the single-fermented wines, especially in terms of esters and aromatic compounds.

### 2.5. Analysis of the Main Flavor Compounds in H. Uvarum–Saccharomyces Cerevisiae Mixed-Fermentation Produced Fruit Wines

Not all volatiles in food products contribute to the aroma. Similarly, volatile compounds with higher concentrations in fruit wines do not necessarily contribute significantly to food aroma. Therefore, OAV is often used to indicate the extent to which a compound contributes to food aroma. The OAV of a compound is related to the compound content in the food and its detection threshold. A volatile aroma compound with an OAV < 1 does not contribute much to the wine aroma, but it positively impacts its harmony and balance. An OAV > 1 suggests a more significant contribution of aroma compounds, which can be perceived by the human olfactory sense and are identified as substances with significant flavor [27].

As shown in Table 5, the aroma compounds with an OAV > 1 in single-fermented, mixed, co-mixed-fermented, and sequentially mixed-fermented fruit wines were primarily esters. Specifically, in *R. roxburghii* fruit wine, the compounds with high OAV values were ethyl acetate, ethyl butyrate, ethyl caproate, ethyl caprylate, and ethyl laurate (Z)-3 hexenol, caprylic acid, and styrene. The highest OAV values were observed in the blueberry wine for ethyl caproate, ethyl caprate, ethyl laurate, 2,4-di-tert-butylphenol, and linalool. These compounds contributed the most to the wine aroma composition and imparted more sweet and fruity aromas. In addition, their OAVs were higher in mixed fermentations than in single fermentations, resulting in more intense and prominent aromas. Esters with OAV >1 included ethyl acetate, ethyl isovalerate, ethyl butyrate, isoamyl acetate, ethyl caproate, ethyl phenylacetate, methyl caprylate, ethyl octanoate, ethyl pelargonate, methyl caprate, ethyl caprate, isoamyl caprylate, ethyl laurate, isoamyl caprate, ethyl myristate, and ethyl palmitate. The OAVs of these esters were generally higher in the mixed-fermented fruit wines. Therefore, mixed fermentation can increase the wine’s ester content and impart a strong floral and fruity aroma. The non-*Saccharomyces cerevisiae* strains significantly positively affected the aroma complexity of the fruit wines. The OAVs of acetic acid were < 1 in all wines across all five different fermentation methods. Excess acetic acid can introduce discordant and irritating odors to the wine, whereas a balanced acetic acid content can reduce the undesirable flavors in the wine. Among the aromatic compounds, 2,4-di-tert-butylphenol had the highest OAV in the blueberry fruit wine, providing more specific aromas. Therefore, wine aroma is not the result of a single aroma compound but rather of all the aromatic compounds working harmoniously, providing pleasant and elegant sensory characteristics.

## 3. Material and Methods

### 3.1. Strain Sources

*S. cerevisiae* (ZYMAFLORE X16) and *H. uvarum* 32349 were purchased from LAFFORT (France) and China Industrial Strain Conservation Center, respectively. The *H. uvarum* F119 strain was isolated from the spontaneous fermentation broth of *R. roxburghii*. The strains X16, 32349, and F119 were streak-cultivated in YPD solid medium, incubated at 28 °C for 48 h in inverted mode, and stored at 4 °C for subsequent use.

### 3.2. Laboratory-Scale Fermentation of *R. roxburghii*, Blueberry, and Plum Fruit Wines

Fresh *R. roxburghii*, blueberry, and plum fruits were purchased from local fruit supermarkets. Subsequently, fresh, ripe, mold-free *R. roxburghii*, blueberry, and plum fruits were crushed and juiced. They were then subjected to enzymolysis with 50 mg/L of pectinase and 50 mg/L of potassium metabisulfite at room temperature for 24 h. Afterwards, the sugar degree of each fruit juice was adjusted to 24 °Brix. The juice of each fruit was divided into five groups: *S. cerevisiae* X16, *H. uvarum* F119-S, *H. uvarum* 32349-S, *H. uvarum* F119-G, and *H. uvarum* 32349-G. *S. cerevisiae* X16: inoculated with *S. cerevisiae* X16 alone at a final concentration of 10^7^ cfu/mL; *H. uvarum* F119-S: inoculated with *H. uvarum* F119 (10^8^ cfu/mL) first, followed by *S. cerevisiae* X16 (10^7^ cfu/mL) after a seven-day fermentation; *H. uvarum* 32349-S: inoculated with *H. uvarum* 32349 (10^8^ cfu/mL) first, followed by *S. cerevisiae* X16 (10^7^ cfu/mL) after a seven-day fermentation; *H. uvarum* F119-G: simultaneous inoculation of *H. uvarum* F119 (10^8^ cfu/mL) and *S. cerevisiae* X16 (10^7^ cfu/mL); *H. uvarum* F119-G: simultaneous inoculation of *H. uvarum* F119-G (10^8^ cfu/mL) and *S. cerevisiae* X16 (10^7^ cfu/mL). The fermentation was carried out at 18 °C, which was maintained constant throughout the different treatments.

### 3.3. Alcohol Content Measurements

The alcohol content, total sugar, total acid, and volatile acid content of *R. roxburghii*, blueberry, and plum fruit wines were measured according to the method of Liu et al. [19].

### 3.4. Measurement of Volatile Components of Fruit Wines

#### 3.4.1. HS–SPME Conditions

Firstly, 8 mL of fruit wine sample was placed in a 20 mL headspace vial containing 1 g of NaCl and 50 L of cyclohexanone (internal standard). The HS–SPME conditions were as follows: extraction head, DVB/C-WR/PDMS (50/30 m, 1 cm); chromatographic column, InertCap Wax capillary column (60 m × 0.25 mm, 0.25 μm); extraction conditions, equilibration at 40 °C for 15 min, extraction at 40 °C for 30 min, and desorption at 240 °C for 2 min.

#### 3.4.2. GC–MS Conditions

The GC conditions were as follows: chromatographic column, InertCap Wax capillary column (60 m × 0.25 mm, 0.25 μm); temperature settings: 40 °C for 3 min, ramped up to 230 °C at 3 °C/min, and held for 2 min; carrier gas (He) flow rate, 1.88 mL/min; injection temperature, 240 °C.

The MS conditions were as follows: electron bombardment ion source; electron energy, 70 eV; ion source temperature, 230 °C; mass spectrometry interface temperature, 250 °C; mass scan range: 29~500 *m*/*z*.

#### 3.4.3. Qualitative and Quantitative Analyses

Qualitative analysis: After the GC–MS analysis, the recovered compounds with more than 80% similarity were annotated using the National Institute of Standards and Technology (NIST) spectral library for preliminary volatile-compound characterization. The linear retention index (LRI) was used for further characterization. Additionally, the volatile compounds in blueberry juice were determined based on the relevant literature [30]. The LRI was calculated according to Equation (1).
(1)$$\;\mathrm{LRI}\;=100\times \left(n+\frac{t_{i}-t_{n}}{t_{n+1}-t_{n}}\right)$$
where *n* is the carbon number of n-alkane, *t_i_* is the retention time of the target compound (min), *t_n_* is the retention time of *C_n_* (min), and *t_n_*_+1_ is the retention time of *C_n_*_+1_ (min).

Quantitative analysis: The volatile compound content was quantified relative to cyclohexanone, which was used as an internal standard. Therefore, the mass concentration of each identified compound could be calculated by the relationship between the mass concentration of the internal standard and the peak area, according to Equation (2).
(2)$$\rho_{i}=\frac{A_{i}}{A_{s}}\times \rho_{s}$$
where *A*_s_ is the peak area of the internal standard, *A*_s_ is the peak area of the unknown compound, *ρ*_i_ is the mass concentration of the internal standard (μg/L), and *ρ*_i_ is the mass concentration of the identified compound (μg/L).

#### 3.4.4. OAV Calculation

The OAV value, the ratio of a volatile compound’s mass concentration to that compound’s threshold value, is primarily used to evaluate the contribution of volatile compounds to the fruit wine odor [6,7]. In general, volatile compounds with an OAV ≥ 1 contribute positively to the aroma of the wine [8]. The OAVs of the fruit wines were calculated according to Equation (3).
(3)$$\mathrm{OAV}=\frac{\rho_{i}}{OT_{i}}$$
where *ρ*_i_ is the mass concentration of the compound (μg/L), and *OT*_i_ is the threshold value of the compound in water (μg/L).

### 3.5. Statistical Analysis

All experiments were replicated three times. The GC–MS data were processed using MS Office 2020, and statistically significant differences were analyzed by SPSS 21.0. *p* < 0.05 indicated a significant difference. Plots were generated using the Origin 2018 software. In addition, the principal component analysis (PCA) was performed using Simca 14.1 (Stockholm, Sweden).

## 4. Conclusions

This is the first study to systematically analyze the effects of *H. uvarum*–*Saccharomyces cerevisiae* mixed fermentation on the aroma characteristics of *R. roxburghii* wine, blueberry, and plum wines. *H. uvarum*–*S. cerevisiae* mixed fermentation contributed to the flavor compounds enrichment of *R. roxburghii*, blueberry, and plum wines. The flavor characteristics of sequentially inoculated fruit wines were quite similar but with certain distinct differences to those of the fruit wines fermented with *S. cerevisiae* alone. Sequential inoculation of *H. uvarum* and *S. cerevisiae* improved the mellowness of the wine and achieved overall better harmony. Therefore, *H. uvarum*–*S. cerevisiae* mixed fermentation can improve the aromatic composition complexity of the wines and empower them with a unique style. Notably, the blueberry wine produced with *H. uvarum*–*Saccharomyces cerevisiae* mixed fermentation had the widest variety and aroma compound content among the fermented wines. This study is the first comparative evaluation of the aroma characteristics of different fruit wines fermented with a mixed inoculation of *H. uvarum* and *S. cerevisiae* and provides a preliminary guide for these fruit wines produced with non-*Saccharomyces* yeast. However, other oenological parameters, including residual sugar, total and volatile acidity, and sensory analysis were also indispensable to fully understand the oenological activities of *H. uvarum* and *S. cerevisiae* in the fermentation of these three kinds of fruit wines.

## Figures and Tables

**Figure 1 molecules-27-08097-f001:**
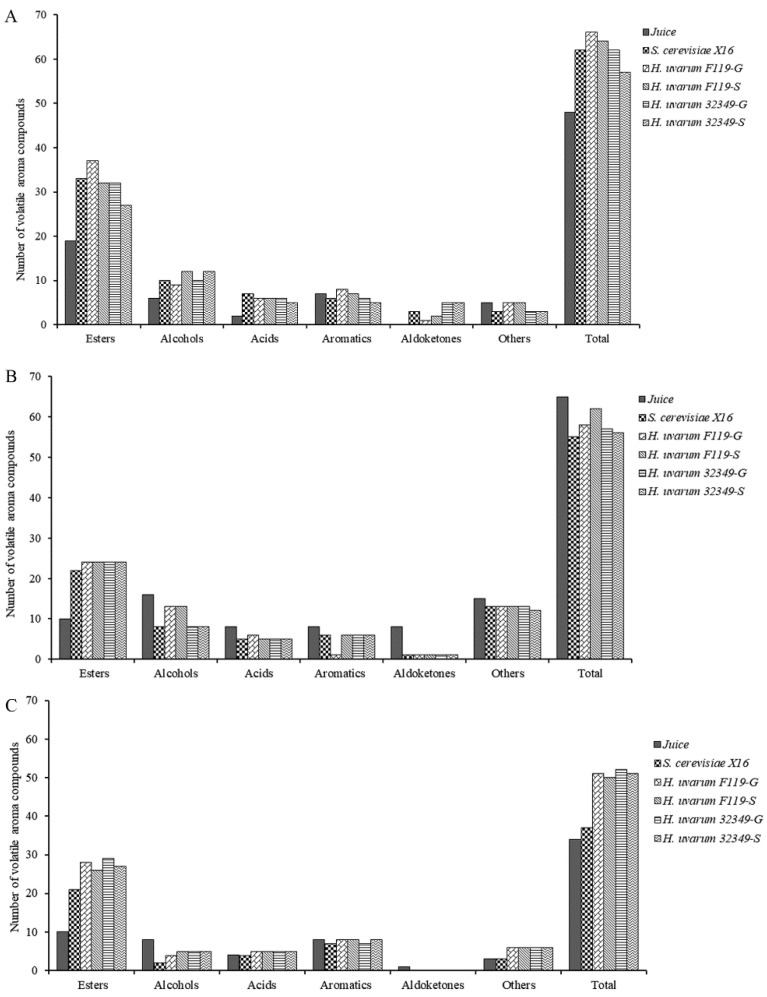
Types of volatile aroma compounds in juices and fermented wines of *R. roxburghii*, blueberry, and plum. (**A**) *R. roxburghii* juice and *R. roxburghii* fruit wine. (**B**) Blueberry juice and blueberry fruit wine. (**C**) Plum juice and plum fruit wine.

**Figure 2 molecules-27-08097-f002:**
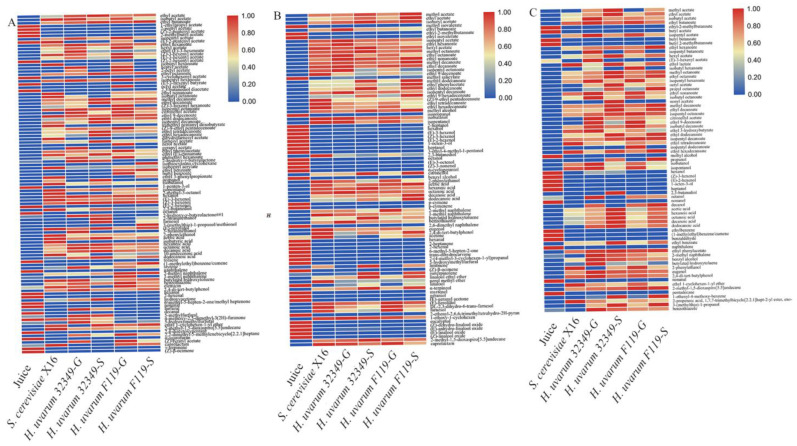
Heatmap of *R. roxburghii* wine, blueberry wine, and plum wine. (**A**) *R. roxburghii* wine; (**B**) blueberry wine; (**C**) plum wine.

**Figure 3 molecules-27-08097-f003:**
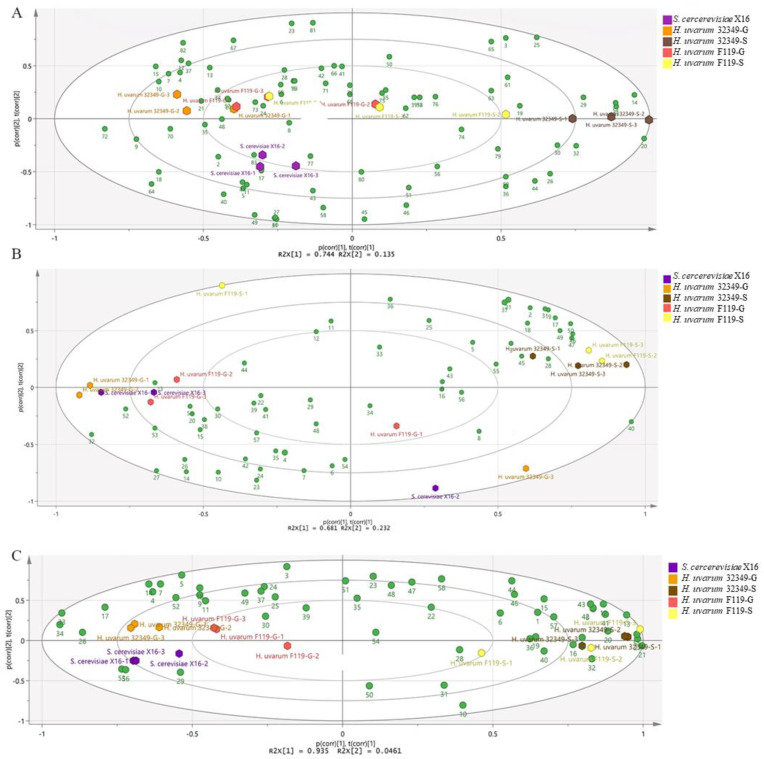
PCA of the wine samples’ volatile compounds in different fermentations. (**A**) *R. roxburghii* wine; (**B**) blueberry wine; (**C**) plum wine.

**Table 1 molecules-27-08097-t001:** Fruit wines alcohol content using different fermentation methods.

Groups	Wines		
	*R. roxburghii*	Blueberry	Kongxinli
*S. cerevisiae* X16	9.67 ± 0.58 ^a^	14.03 ± 0.06 ^c^	17.57 ± 0.51 ^b^
*H. uvarum* F119-G	9.40 ± 0.36 ^a^	13.00 ± 0.00 ^b^	16.00 ± 0.00 ^a^
*H. uvarum* F119-S	9.83 ± 0.29 ^a^	12.00 ± 0.00 ^a^	17.6 ± 0.50 ^b^
*H. uvarum* 32349-G	9.00 ± 0.50 ^a^	13.00 ± 0.00 ^b^	15.47 ± 0.40 ^a^
*H. uvarum* 32349-S	10.17 ± 0.58 ^a^	12.17 ± 0.29 ^a^	17.00 ± 0.00 ^b^

Note: Values in the same column with different lowercase letters are significantly different (*p* < 0.05).

**Table 2 molecules-27-08097-t002:** Volatile aroma compound content (mg/L) in *R. roxburghii* juice and *R. roxburghii* fruit wines.

	Juice	*S. cerevisiae* X16	*H. uvarum* F119-G	*H. uvarum* F119-S	*H. uvarum* 32349-G	*H. uvarum* 32349-S
Esters	16.88 ± 0.75 ^a^	1454.12 ± 19.01 ^b^	1699.73 ± 93.79 ^c^	1715.82 ± 122.56 ^c^	1561.02 ± 94.02 ^c^	1996.10 ± 32.96 ^d^
Alcohols	3.77 ± 0.13 ^a^	232.58 ± 5.73 ^d^	163.70 ± 12.66 ^b^	170.38 ± 2.75 ^b^	161.82 ± 8.76 ^b^	193.52 ± 4.79 ^c^
Acids	1.41 ± 0.48 ^a^	86.85 ± 8.08 ^b^	81.33 ± 20.28 ^b^	90.76 ± 20.86 ^b^	102.65 ± 15.12 ^b^	111.44 ± 6.71 ^b^
Aromatics	2.83 ± 0.21 ^a^	128.01 ± 80.03 ^b^	107.83 ± 11.58 ^b^	118.02 ± 53.98 ^b^	138.10 ± 16.57 ^b^	57.28 ± 51.87 ^ab^
Aldoketones	4.84 ± 0.28 ^b^	2.14 ± 0.27 ^a^	1.28 ± 0.12 ^a^	2.49 ± 0.33 ^a^	13.60 ± 2.15 d	10.11 ± 1.00 ^c^
Others	1.15 ± 0.05 ^a^	45.01 ± 16.47 ^b^	32.57 ± 7.96 ^b^	32.57 ± 7.94 ^b^	34.73 ± 8.07 ^b^	34.71 ± 10.45 ^b^
Total	30.88 ± 1.91 ^a^	1948.69 ± 129.59 ^b^	2086.45 ± 146.38 ^b^	2130.04 ± 208.4 ^b^	2011.92 ± 144.69 ^b^	2403.16 ± 107.78 ^b^

Note: Values in the same column with different lowercase letters are significantly different (*p* < 0.05).

**Table 3 molecules-27-08097-t003:** Volatile aroma compound content (mg/L) in blueberry juice and blueberry fruit wines.

	Juice	*S. cerevisiae* X16	*H. uvarum* F119-G	*H. uvarum* F119-S	*H. uvarum* 32349-G	*H. uvarum* 32349-S
Esters	5.29 ± 0.24 ^a^	933.40 ± 59.46 ^b^	924.83 ± 47.58 ^b^	1005.49 ± 85.51 ^b^	1013.06 ± 25.40 ^b^	998.53 ± 38.97 ^b^
Alcohols	22.18 ± 0.08 ^a^	333.37 ± 4.10 ^c^	307.16 ± 28.74 ^c^	213.53 ± 13.41 ^b^	314.96 ± 10.80 ^c^	212.03 ± 5.65 ^b^
Acids	0.99 ± 0.39 ^a^	55.34 ± 7.28 ^bc^	56.12 ± 4.45 ^bc^	65.01 ± 2.09 c	47.66 ± 6.86 ^b^	56.35 ± 8.02 bc
Aromatics	10.11 ± 0.59 ^a^	125.10 ± 120.44 a	113.69 ± 77.49 ^a^	180.18 ± 127.31 ^a^	118.83 ± 153.92 ^a^	242.28 ± 28.32 ^a^
Aldoketones	1.54 ± 0.05 ^b^	0.22 ± 0.04 ^a^	0.21 ± 0.02 ^a^	0.22 ± 0.01 ^a^	0.20 ± 0.03 ^a^	0.19 ± 0.04 ^a^
Others	38.69 ± 1.45 ^a^	51.32 ± 5.39 ^b^	50.48 ± 6.24 ^b^	57.81 ± 2.02 ^bc^	55.30 ± 6.70 ^bc^	63.82 ± 1.69 ^c^
Total	78.80 ± 2.80 ^a^	1498.75 ± 196.71 ^b^	1452.49 ± 164.52 ^b^	1522.24 ± 230.35 ^b^	1551.01 ± 203.71 ^b^	1573.20 ± 82.69 ^b^

Note: Values in the same column with different lowercase letters are significantly different (*p* < 0.05).

**Table 4 molecules-27-08097-t004:** Volatile aroma compound content (mg/L) in plum juice and plum fruit wines.

	Juice	*S. cerevisiae* X16	*H. uvarum* F119-G	*H. uvarum* F119-S	*H. uvarum* 32349-G	*H. uvarum* 32349-S
Esters	3.15 ± 0.08 ^a^	2455.52 ± 84.25 ^b^	2943.59 ± 16.12 ^d^	3657.07 ± 228.17 ^e^	2744.55 ± 23.82 ^c^	3777.30 ± 94.59 ^e^
Alcohols	3.34 ± 0.04 ^a^	321.90 ± 0.94 ^d^	337.91 ± 2.45 ^e^	182.72 ± 12.26 ^b^	357.56 ± 1.39 ^f^	200.80 ± 6.06 ^c^
Acids	1.62 ± 0.32 ^a^	30.24 ± 1.59 ^b^	41.10 ± 2.94 ^c^	64.90 ± 8.01 ^e^	53.13 ± 3.07 ^d^	74.40 ± 3.97 ^f^
Aromatics	2.54 ± 0.09 ^a^	63.12 ± 14.80 ^ab^	87.36 ± 42.91 ^b^	56.55 ± 15.54 ^ab^	112.03 ± 54.31 ^b^	73.14 ± 18.21 ^ab^
Aldoketones	0.25 ± 0.05 ^b^	nd	nd	nd	nd	nd
Others	0.62 ± 0.17 ^a^	11.66 ± 2.86 ^b^	15.90 ± 4.00 ^b^	19.15 ± 5.04 ^b^	14.21 ± 0.99 ^b^	17.84 ± 3.29 ^b^
Total	11.52 ± 0.75 ^a^	2884.44 ± 104.44 ^b^	3425.86 ± 68.42 ^c^	3980.39 ± 269.02 ^d^	3281.48 ± 29.28 ^c^	4143.48 ± 126.12 ^d^

Note: Values in the same column with different lowercase letters are significantly different (*p* < 0.05). nd represents a compound that is not detected.

**Table 5 molecules-27-08097-t005:** Volatile flavor compounds across the different fermentation methods.

	NO.	Compounds	Odor Quality [28,29]	Odor Threshold		OAV				
				(mg/L)	Juice	*S. cerevisiae* X16	*H. uvarum* F119-G	*H. uvarum* F119-S	*H. uvarum* 32349-G	*H. uvarum* 32349-S
*R. roxburghii* wine	1	Ethyl acetate	Sweet, pineapple	7.50	231	3	7	7	8	9
	2	Ethyl butanoate	Sweet, pineapple	0.02	23231	111	148	162	142	159
	3	Isopentyl acetate	Sweet, banana	0.03	2766	3391	4757	3179	4520	3176
	4	Ethyl hexanoate	Sweet, fruity	0.005	49,039	20,792	23,470	21,942	25,874	20,836
	5	Hexyl acetate	Apple, banana	0.67	63	43	75	42	67	35
	6	Ethyl octanoate	Banana, brandy	0.002	nd	227,870	246,960	266,630	229,475	299,190
	7	Ethyl decanoate	Apple, grape	0.20	nd	2599	2415	2743	2059	3679
	8	Isopentyl octanoate	Pineapple, coconut	0.125	nd	25	27	23	27	24
	9	Ethyl 9-decenoate	Fruity, fatty	0.1	nd	429	428	390	393	374
	10	Ethyl phenylacetate	Floral, honey	0.25	nd	80	97	55	78	64
	11	Ethyl dodecanoate	Sweet, creamy	0.5	nd	160	410	460	358	519
	12	Ethyl tetradecanoate	Sweet, creamy	0.5	nd	3	4	4	4	6
	13	Ethyl (E)-cinnamate	Floral, honey	0.001	nd	590	390	380	390	630
	14	Isopentanol	Fermented, alcohol	30	5	5	3	4	3	4
	15	Hexanol	Fruity	8	208	1	1	1	1	1
	16	(E)-3-Hexenol	Green leaf, fruity	1	0	12	9	9	9	9
	17	(Z)-3-Hexenol	Grassy, herbaceous	0.4	716	27	30	30	30	30
	18	2-Phenylethanol	Sweet, rose	10	nd	5	3	2	3	3
	19	Acetic acid	Sour, vinegar	200	6	<1	<1	<1	<1	<1
	20	Hexanoic acid	Sour, sweaty	3	30	1	1	1	1	1
	21	Octanoic acid	Fatty, sour	0.5	277	41	58	47	63	57
	22	Decanoic acid	Fatty, sour	15	7	1	2	2	2	2
	23	Dodecanoic acid	Fatty, coconut	1	nd	1	4	6	7	7
	24	Styrene	Sweet, floral	0.125	nd	9	29	21	38	19
	25	4-Methoxy-2,5-dimethyl-3(2H)-furanone	Sweet, caramel	0.016	nd	89	79	78	89	85
Blueberry wine	1	Ethyl acetate	Sweet, pineapple	7.5	nd	12	15	23	15	20
	2	Ethyl butyrate	Fruity, pineapple	0.02	<1	67	61	47	98	66
	3	Ethyl 2-methylbutyrate	Sweet, green	0.018	14	nd	nd	nd	nd	nd
	4	Ethyl isovalerate	Fruity, sweet	0.003	762	66	50	83	75	74
	5	Isoamyl acetate	Sweet, fruity	1.6	<1	11	11	10	13	11
	6	Ethyl Hexanoate	Sweet, fruity	0.014	4	3381	3 136	2367	4011	3020
	7	Hexyl acetate	Fruity, green	0.67	<1	2	2	2	2	2
	8	Ethyl caprylate	Fruity, wine	0.58	<1	365	343	235	405	287
	9	Ethyl caprate	Sweet, waxy	0.2	<1	1676	1644	1634	1777	1442
	10	Fema 2080	Sweet, oily	0.125	6	13	11	7	14	8
	11	Methyl salicylate	Sweet, mint	0.1	12	6	6	6	7	6
	12	Ethyl phenylacetate	Sweet, floral	0.073	<1	77	60	177	80	127
	13	Ethyl laurate	Sweet, waxy	1.5	<1	110	118	183	120	195
	14	Ethyl myristate	Sweet, waxy	2	<1	2	2	1	2	1
	15	Palmitic acid ethyl ester	Waxy, fruity	1.5	<1	6	6	3	5	4
	16	Methanol	Alcoholic	0.1	7	24	24	27	22	26
	17	1-Hexanol	Fruity sweet, green	8	2	<1	<1	<1	<1	<1
	18	1-Octen-3-ol	Fruity, sweet, green	0.02	7	nd	nd	nd	nd	nd
	19	Citronellol	Floral, waxy	0.01	2	13	12	20	13	21
	20	Phenyl ethanol		14	<1	6	5	3	5	2
	21	Hexanoic acid	Sour, fatty, sweet	0.42	<1	3	3	3	3	4
	22	Octanoic acid	Fatty, waxy	0.5	<1	13	13	12	14	13
	23	Decanoic acid	Sour, fatty	1	<1	8	9	14	10	10
	24	Lauric acid	Mild, fatty	1	<1	1	1	4	1	3
	25	Benzothiazole	Mild, fatty	0.08	2	6	6	5	8	6
	26	Eugenol	Sweet, spicy	0.006	19	62	56	46	51	50
	27	2,4-Di-tert-butylphenol	Phenolic	0.036	254	334	299	331	279	315
	28	Hexanal	Fresh, green	0.005	73	nd	nd	nd	nd	nd
	29	6-Methyl-5-hepten-2-one	Green, musty	0.05	2	nd	nd	nd	nd	nd
	30	Linalool	Floral, sweet	0.015	1805	1188	1202	1454	1325	1490
	31	Alpha-Terpineol	Pine, woody	0.25	8	14	14	17	15	17
	32	Geraniol	Sweet, floral	0.03	51	nd	nd	nd	nd	nd
Plum wine	1	Ethyl acetate	Sweet, pineapple	7.5	<1	7	10	13	11	16
	2	Isobutyl acetate	Sweet, banana	0.03	nd	76	116	101	146	106
	3	Ethyl butanoate	Sweet, pineapple	0.02	23	443	489	413	509	440
	4	Butyl acetate	Sweet, banana	0.01	12	nd	nd	nd	nd	nd
	5	Butyl butanoate	Banana, pineapple	0.0028	28	nd	nd	nd	nd	nd
	6	Butyl 2-methylbutanoate		0.017	6	nd	nd	nd	nd	nd
	7	Ethyl hexanoate	Sweet, fruity	0.014	18	5770	7685	10,005	7956	8402
	8	Isopentyl butanoate	Pineapple, pear	0.015	nd	nd	15	nd	18	
	9	Hexyl acetate	Apple, banana	0.67	<1	2	2	3	3	4
	10	Methyl octanoate	Sweet, orange wine	0.2	nd	7	9	11	9	12
	11	Ethyl octanoate	Banana, brandy	0.0193	nd	37,762	45,720	56,984	44,064	58,757
	12	Isopentyl hexanoate	Banana, apple	0.32	nd	5	7	4	7	5
	13	Octyl acetate	Floral, herbal, fruity	0.047	nd		6	8	7	11
	14	Ethyl nonanoate	Rose, rum	0.39	nd	14	17	11	17	11
	15	Ethyl decanoate	Apple, grape, fatty	0.005	nd	221,642	240,867	333,258	209,383	346,736
	16	Isopentyl octanoate	Floral, fresh, fruity	0.07	nd	152	218	194	189	206
	17	Citronellyl acetate	Rose, orange, honey	1	nd	nd	1	1	2	1
	18	Ethyl dodecanoate	Sweet, creamy	1.5	nd	128	158	230	131	233
	19	Ethyl tetradecanoate	Sweet, creamy	4	nd	2	1	1	1	1
	20	Ethyl hexadecanoate	Waxy, fruity	1.5	nd	5	4	5	3	5
	21	Methyl alcohol	Alcoholic	0.1	2	nd	nd	nd	nd	nd
	22	Isopentanol	Fermented, alcohol	30	<1	10	10	5	11	6
	23	1-Octen-3-ol	Mushroom, earthy	0.0015	72	nd	nd	nd	nd	nd
	24	Heptanol	Fresh, herbal	0.0054	18	nd	nd	nd	nd	nd
	25	Octanol	Waxy, orange	0.01258	2	nd	4	4	5	nd
	26	Nonanol	Floral, fresh, fatty	0.0455	10	nd	nd	nd	nd	nd
	27	Octanoic acid	Fatty, sour	0.5	<1	21	30	52	43	60
	28	Decanoic acid	Fatty, sour	15	<1	<1	1	2	1	2
	29	Benzaldehyde	Sweet, bitter, Cherry	0.35	2	nd	nd	nd	nd	nd
	30	Ethyl benzoate	Floral, fruity	0.053	nd	36	33	54	36	65
	31	2-Phenylethanol	Sweet, rose	0.14	1	204	197	123	nd	136
	32	Eugenol	Sweet, woody	0.00071	nd	4869	5942	5510	6755	6117
	33	2,4-Di-tert-butyl-phenol		0.5	1	20	21	18	23	20
	34	Nonanal	Waxy, rose, fresh	0.0011	229	nd	nd	nd	nd	nd
	35	Benzothiazole	Sulfuric, vegetable	0.08	1	nd	7	6	9	7

Note: nd represents a compound that is not detected.

## Data Availability

Not applicable.

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
