# Peer review of "Effect of Hanseniaspora uvarum–Saccharomyces cerevisiae Mixed Fermentation on Aroma Characteristics of Rosa roxburghii Tratt, Blueberry, and Plum Wines"

_molecules, 2022, doi:10.3390/molecules27228097_

Round 1
Reviewer 1 Report
Molecules-2003478
The review of the manuscript entitled “Effect of Hanseniaspora uvarum–Saccharomyces cerevisiae mixed-fermentation on aroma characteristics of Rosa roxburghii Tratt, blueberry, and plum wines”. This article concerns a the use of fresh Rosa roxburghii Tratt (R.roxburghii), blueberry, and plum as the main brewing ingredients. Then H. uvarum and S. cerevisiae were used as the mixed-fermentation agents. As a sample preparation method the headspace solid phase microextraction (HS–SPME) combined with gas chromatography-mass spectrometry (GC–MS) were applied to analyze effects of fermentation process on volatiles in fruit wines. Also, volatile components' contribution to the different fruit wine aromas was estimated using odor active value (OAV).
Some topics of these invstigations are interesting, but the relevance of the study in general has not been explained. The question is what is the originality of this review study and their scientific significance. Moreover, mentioned investigations more specifically the evaluation of data available in databases, is not particularly original and several major improvements are still needed in order to achieve a relevant manuscript. After reading this manuscript, some comments and qualifications came to mind as follows.
· Abstract, there is no significant information. A big mistake is the lack of sufficient data about sufficient results of quantitative analysis and conclusions, new approach in the field of improving the fruit wine brewing processes. The content should change significantly.
· Likewise, in the introduction glaring is the lack of sufficient information and this part requires some revision.
· Chapter Results - the critical evaluation of more important quantitative merits (such as LOD, LOQ, sensitivity, linear range, accuracy, etc) is required. Presented results of raw plant material quantitative analysis are also missing. It should be completed in the manuscript.
· Figure 2. Heatmap of R. roxburghii wine, blueberry wine, and plum wine. A, R.roxburghii wine; B, blueberry wine; C, Plum wine - it is incomplete and illegible.
· Any scientific work should contain important elements of novelty and a new approach to the discussed problems. This aspect is certainly missing from the reviewed work. What are the significant differences from other scientific works in this field? Moreover, please identify advantages and disadvantages of the proposed methodology, discuss the possibilities of the obtained results use and what are the outlooks for the future?
Summary:
The subject of the manuscript falls within the scope of Molecules. Nevertheless, due to the lack of essential elements of scientific work, the reviewed manuscript is recommended to publish after major revision.
Author Response
Dear Editor:
This is a revision of our previous manuscript (Molecules-2003478). First, we thank the reviewers for their critical comments and constructive suggestions. Based on the suggestions, we performed many revisions of this manuscript by using tracked changes. For the response to reviewers, the comments are black, author's answers are blue.
Sincerely yours,
Professor Liu
Guizhou Institute of Technology, Guiyang, China

Reviewer 2 Report
The paper “Effect of Hanseniaspora uvarum–Saccharomyces cerevisiae mixed-fermentation on aroma characteristics of Rosa roxburghii Tratt, blueberry, and plum wines” contributes to the growth of literature for research on food, especially the analysis of aromatic compounds and sensory analysis. The paper should be interesting for producers who offer food to consumers, especially fruit vines consumers.
Before the manuscript acceptation for publication in “Molecules” the following items should be revised:
Abstract
The authors did not show sample values
Introduction
The description of the aim of the paper is not specific.
Material and Methods
The " Material and Methods” section should follow after the Discussion.
Line 114-115 “Additionally, the volatile compounds in blueberry juice were determined based on the relevant literature.” The authors did not provide the literature.
Figure 1.
There are no results for the significance of differences. I suggest you show graphs with the significance of the differences
The maximum values of the vertical axes are different
Figure 2 and Figure 3.
the graphics and description are hardly visible
Line 422-423
“…ore significant contribution of aroma compounds, which can be perceived by the human olfactory sense and are identified as significant flavor substances [27].”
- positive or negative?
Table 5 “Odor quality”, - on what basis, i.e. Literature?
The discussion
The authors did not describe the discussion of the results
Conclusions
Line 453 “enrichment of the flavor characteristics of…” In which direction of enrichment?
Is this based on research among consumers or are they assumptions? Did the authors mean: increasing the content of these compounds?
The authors should add the summary conclusion - the positive or negative effects of the research. What are the limitations of this research?
Author Response

(The authors gave the same response as above.)

Round 2
Reviewer 1 Report
Review of manuscript number Molecules-2003478_R2
The revised manuscript in some parts has been improved according to my comments, but not enough in some points. Nevertheless, I think that the manuscript in its present form can be published.